# COVID-19 in China and the US: Differences in Hospital Admission Co-Variates and Outcomes

**DOI:** 10.3390/vaccines10020326

**Published:** 2022-02-18

**Authors:** Yulin Cao, Di Wu, Kuo Zeng, Lei Chen, Jianming Yu, Wenjuan He, Li Chen, Wenxiang Ren, Fei Gao, Wenlan Chen, Hongxiang Wang, Robert Peter Gale, Zhichao Chen, Qiubai Li

**Affiliations:** 1Institute of Hematology, Union Hospital, Tongji Medical College, Huazhong University of Science and Technology, Wuhan 430022, China; yulincao123@outlook.com (Y.C.); doctortian0209@163.com (D.W.); lene918@163.com (L.C.); xiaoyusteve@hotmail.com (J.Y.); hwjsun@163.com (W.H.); rwx15093376011@outlook.com (W.R.); gaof0301@126.com (F.G.); wenlanchen@yahoo.com (W.C.); chenzhichao@hust.edu.cn (Z.C.); 2Department of Neurosurgery, Renmin Hospital of Wuhan University, Wuhan 430060, China; zengkuo1990@gmail.com; 3Department of Hematology, Wuhan Central Hospital, Tongji Medical College, Huazhong University of Science and Technology, Wuhan 430014, China; lili2002311@163.com (L.C.); wanghongxiang_wh@163.com (H.W.); 4Center for Hematology Research, Department of Immunology and Inflammation, Imperial College London, London W12 0NM, UK; robertpetergale@gmail.com; 5Hubei Engineering Research Center for Application of Extracelluar Vesicles, Hubei University of Science and Technology, Xianning 437100, China

**Keywords:** COVID-19, hospital admission co-variates, outcomes

## Abstract

(1) Background: Although there are extensive data on admission co-variates and outcomes of persons with coronavirus infectious disease-2019 (COVID-19) at diverse geographic sites, there are few, if any, subject-level comparisons between sites in regions and countries. We investigated differences in hospital admission co-variates and outcomes of hospitalized people with COVID-19 between Wuhan City, China and the New York City region, USA. (2) Methods: We retrospectively analyzed clinical data on 1859 hospitalized subjects with COVID-19 in Wuhan City, China, from 20 January to 4 April 2020. Data on 5700 hospitalized subjects with COVID-19 in the New York City region, USA, from 1 March to 4 April 2020 were extracted from an article by Richardson et al. Hospital admission co-variates (epidemiological, demographic, and laboratory co-variates) and outcomes (rate of intensive care unit [ICU] admission, invasive mechanical ventilation [IMV], major organ failure and death, and length of hospital stay) were compared between the cohorts. (3) Results: Wuhan subjects were younger, more likely female, less likely to have co-morbidities and fever, more likely to have a blood lymphocyte concentration > 1 × 10^9^/L, and less likely to have abnormal liver and cardiac function tests compared with New York subjects. There were outcomes data on all Wuhan subjects and 2634 New York subjects. Wuhan subjects had higher blood nadir median lymphocyte concentrations and longer hospitalizations, and were less likely to receive IMV, ICU hospitalization, and interventions for kidney failure. Amongst subjects not receiving IMV, those in Wuhan were less likely to die compared with New York subjects. In contrast, risk of death was similar in subjects receiving IMV at both sites. (4) Conclusions: We found different hospital admission co-variates and outcomes between hospitalized persons with COVID-19 between Wuhan City and the New York region, which should be useful developing a comprehensive global understanding of the SARS-CoV-2 pandemic and COVID-19.

## 1. Introduction

The severe acute respiratory syndrome coronavirus-2 (SARS-CoV-2) pandemic, which causes coronavirus infectious disease-2019 (COVID-19), began in Wuhan City, China, in December 2019. By 1 March 2020 the pandemic had spread to the New York City region. Molecular studies indicate SARS-CoV-2 mutated substantially in this interval [1]. Whether this affected infectivity and/or virulence is unknown [2,3].

Several studies report risk factors for SARS-CoV-2-infection and for developing COVID-19. For example, older age, male sex, cancer, and chronic diseases. There are also considerable data regarding risk factors for death from COVID-19, such as older age; male sex; co-morbidities, such as arterio-sclerotic cardio- and vascular disease (ASCVD); chronic obstructive pulmonary disease (COPD); diabetes and cancer; and abnormal laboratory covariates, including high D-dimmer concentration, high neutrophil-to-lymphocyte ratio, low blood platelet concentration, high procalcitonin concentration, and increased interleukin-6 concentration [4,5,6]. In addition to these risk factors there are substantial differences in diagnostic criteria for COVID-19 and for hospital admission between different geographic regions, countries, and cities [7,8,9,10].

We interrogated data on hospital admission co-variates and outcomes of persons with COVID-19 in Wuhan City, China, and the New York City region, and detected substantial differences. These data should be useful in developing a comprehensive global understanding of the SARS-CoV-2 pandemic and COVID-19.

## 2. Materials and Methods

For the Wuhan cohort, the following details of methods are consistent with those in our previous publications [9,10,11]. For the New York region cohort, we used data reported by Richardson et al. [12].

### 2.1. Subjects

In total, 1859 consecutive persons ≥18 years 20 January to 4 April 2020 from Union Hospital (main part, Union West Hospital and Union Tumor Hospital), Wuhan Central Hospital, General Hospital of Central Theater Command, PLA, Wuhan Third Hospital, and Wuhan Jin-Yin-Tan Hospital were studied. These hospitals were re-constructed and designated as COVID-19 treatment centers. Between 4 February and 18 February 2020, persons with clinical symptoms and a lung computed tomography (CT) scan consistent with COVID-19 were diagnosed as having COVID-19 without confirmation of SARS-CoV-2-infection by quantitative reverse transcript polymerase chain reaction (qRT-PCR). After hospitalization, subjects were tested by qRT-PCR to confirm the diagnosis and monitor their course. Beginning 29 February 2020, anti-SARS-CoV-2 IgM and/or IgG antibodies were assayed at Union Hospital and Wuhan Central Hospital by the centers to confirm the diagnosis and to evaluate suspected cases of COVID-19 which were qRT-PCR-negative. Subjects in whom we could not confirm SARS-CoV-2-infection by a qRT-PCR, IgM/IgG assay or both were excluded from the study. Subjects recovering from COVID-19 were discharged and transferred to designated hotels, Fangcang shelter hospitals, or Leishenshan Hospital for 2–4 weeks of isolation or further care if needed [13].

For the New York cohort, COVID-19 cases confirmed by positive result on PCR testing of a nasopharyngeal sample (the initial test result or if it was negative but repeat testing was positive) and considered to be sufficiently medically ill to require hospital admission were admitted to any of 12 hospitals in Northwell Health, the largest academic health system in New York, between 1 March 2020 and 4 April 2020.

### 2.2. Data Collection

The data collection procedures are routinely followed as described in detail in our previous publications [10]. For the Wuhan cohort, we obtained epidemiological, demographic, clinical, laboratory, radiological, therapy, and outcomes data from electronic medical records (EMRs) using a standardized data collection form. Interventions included antibiotics, anti-viral drugs, corticosteroids, and supportive care, including supplemental oxygen, mechanical ventilation (with and without intubation), and extra-corporeal membrane oxygenation (ECMO). Data were independently entered and cross-validated by two researchers (W.H. and J.Y.). A third researcher (Q.L.) adjudicated discordances. Missing data were retrieved from the relevant hospital.

For the New York cohort, data collected included patient demographic information, comorbidities, home medications, triage vitals, initial laboratory tests, initial electrocardiogram results, diagnoses during the hospital course, inpatient medications, treatments, and outcomes. Data were collected from the enterprise electronic health record (Sunrise Clinical Manager; Allscripts) reporting database.

### 2.3. Definitions

For the Wuhan cohort, exposure history was defined as exposure to persons with confirmed SARS-CoV-2-infection or visiting the Huanan Wholesale Seafood Market, possible origin site of the SARS-CoV-2 epidemic in Wuhan City. Smoking history was defined as current or former smoker (stopping >5 years ago) with exposure of ≥20 cigarettes per day for ≥1 year (1 pack year). Fever was defined as temperature ≥37.3 °C on ≥2 measurement >4 h apart. Acute kidney injury, acute respiratory distress syndrome (ARDS) and acute cardiac injury were diagnosed according to guidelines or as reported. Acute liver damage was defined as an elevation in aspartate aminotransferase or alanine aminotransferase of >15 × upper limit of normal. Severity of COVID-19 was classified as: (1) mild; (2) moderate; (3) severe; or (4) critical, according to the Chinese guideline for COVID-19 (version 7) [14,15,16]. Recovery was defined as complete resolution of all clinical signs and symptoms, normalization of the lung computed tomography (CT) scan (if abnormal) and ≥2 negative quantitative real-time polymerase chain reaction (qRT-PCR) tests for SARS-CoV-2 [17,18]. Subjects dying of unrelated causes were excluded from analyses of COVID-19-related deaths.

For the New York cohort, acute kidney injury was identified as an increase in serum creatinine by 0.3 mg/dL or more (≥26.5 μmol/L) within 48 h or an increase in serum creatinine to 1.5 times or more baseline within the prior 7 days compared with the preceding 1 year of data in acute care medical records based on the Kidney Disease: Improving Global Outcomes (KDIGO) definition. Acute hepatic injury was defined as an elevation in aspartate aminotransferase or alanine aminotransferase of more than 15 times the upper limit of normal.

### 2.4. Statistical Analyses

Demographics and clinical co-variates were presented using descriptive statistics with frequencies (percentage) for discrete variables and median (interquartile range, IQR) and range for continuous variables. Descriptive statistics with frequencies (percentage) for discrete variables were compared using χ^2^ test; a two-sided alpha of <0.05 was considered significant. Median (IQR) and range for continuous variables are displayed. It was not possible to calculate *p*-values for comparisons of continuous co-variates because of limitations of the New York data.

## 3. Results

### 3.1. Comparison of Hospitalized Subjects from Wuhan and New York Cities by Admission Clinical Co-Variates

Subjects in the Wuhan cohort were younger (median 59 years [IQR 45–68 years] versus 63 years [IQR 52–75]; Table 1), more likely female (50%, 925/1859 versus 40%, 2263/5700; *p* < 0.001), and less likely smokers (6%, 111/1859 versus 16%, 558/5700; *p* < 0.001). Wuhan subjects were less likely to have co-morbidities, such as ASCVD (14%, 268/1859 versus 18%, 966/5700; *p* < 0.001), hypertension (31%, 579/1859 versus 57%, 3026/5700; *p* < 0.001), diabetes (14%, 262/1859 versus 34%, 1808/5700; *p* < 0.001), chronic obstructive pulmonary disease (COPD; 3%, 61/1859 versus 5%, 287/5700; *p* < 0.001), cancer (3%, 61/1859 versus 6%, 320/5700; *p* < 0.001), and chronic kidney disease (2%, 45/1859 versus 9%, 454/5700; *p* < 0.001), and to have >1 co-morbidity (*p* < 0.001). Subjects from Wuhan were less likely to have temperature > 38 °C on admission (10%, 189/1859 versus 31%, 1734/5700; *p* < 0.001).

On admission, Wuhan subjects were less likely to have a blood lymphocyte concentration <1 × 10^9^/L (40%, 736/1859 versus 60%, 3387/5700; *p* < 0.001), lower median blood neutrophil concentration (3 × 10^9^/L [IQR 2–5 × 10^9^/L] versus 5.3 × 10^9^/L [IQR 3.7–7.7 × 10^9^/L], alanine aminotransferase (ALT) >60 U/L (29%, 533/1859 versus 39%, 2176/5700; *p* < 0.001), serum aspartate aminotransferase (AST) >40 U/L (35%, 645/1859 versus 58%, 3263/5700; *p* < 0.001), lactate dehydrogenase (LDH) (212 U/L [IQR 170–292 U/L] versus 404 U/L [IQR 300–552 U/L]), troponin-I > upper limit of normal (19%, 203/1859 versus 23%, 801/5700; *p* = 0.006), B-type natriuretic peptide (BNP; 61 pg/mL [IQR 18–242 pg/mL] versus 386 pg/mL, [IQR 106–1997 pg/mL]), creatine kinase (88 U/L [IQR 54–165 U/L] versus 171 U/L [IQR 84–397 U/L]), C-reactive protein (CRP) (13 mg/L [IQR 3–51 mg/L] versus 130 mg/L [IQR 64–269 mg/L], ferritin (567 ng/mL [IQR 246–1218] versus 798 ng/mL [IQR 411–1515 ng/mL]), and procalcitonin (0.06 ng/mL [IQR 0.05–0.1 ng/mL] versus 0.2 ng/ml [IQR 0.1–0.6 ng/mL]). These data are displayed in Table 1.

### 3.2. Comparison of Outcomes by Co-Variates after Admission

There were outcomes data on the 1859 Wuhan subjects and 2634 New York subjects (Table 2). Wuhan subjects had higher blood nadir median lymphocyte concentrations (1.0 × 10^9^/L [0.8–1.6 × 10^9^/L] versus 0.88 × 10^9^/L [0.6–1.2 × 10^9^/L]) and longer hospital stays (18 days [range, 12–23 days] versus 4 [range, 2–7 days]). They were less likely to receive invasive mechanical ventilation (5%, 85/1859 versus 12%, 320/2634; *p* < 0.001), ICU care (6%, 106/1859 versus 14%, 373/2634; *p* < 0.001), to have acute kidney injury (5%, 99/1859 versus 22%, 523/2634; *p* < 0.001) but not acute liver injury (1%, 27/1859 versus 2%, 56/2634; *p* = 0.116), and to receive therapy of kidney failure (1%, 23/1859 versus 3%, 81/2634; *p* < 0.001). Wuhan subjects were less likely to die (11%, 209/1859 versus 21%, 553/2634; *p* < 0.001). In subjects not receiving mechanical ventilation, those in Wuhan were less likely to die compared with those in New York (8%, 136/1774 versus 12%, 271/2314; *p* < 0.001). In contrast, death rates were similar in subjects in both cities in subjects receiving mechanical ventilation (86%, 73/85 versus 88%, 282/320; *p* = 0.58).

### 3.3. Comparison of Deaths by Sex and 10-Year Intervals of Age

In the Wuhan cohort, males aged 40–49, 80–89, and ≥90 years were less likely to die compared with New York males of a similar age, whereas those >49 years to <80 years had similar risks of death (Table 3). In the Wuhan cohort, females of all ages were less likely to die compared with New York. In subjects who died, those in Wuhan had longer median hospital stays compared with those in New York (Table 3).

## 4. Discussion

We compared hospital admission co-variates and outcomes of persons with COVID-19 between Wuhan City, China, and the New York City region, two of the epicenters of the SARS-CoV-2 pandemic. We found many differences in baseline co-variates, such as sex, age, co-morbidities, and laboratory parameters associated and unassociated with COVID-19 [19,20]. There were also significant differences in outcomes. For example, Wuhan subjects were more likely female, younger and have fewer co-morbidities including ASCVD, hypertension, diabetes mellitus, COPD, and kidney failure compared with New York subjects.

There are several possible reasons for these discordances. One could be that people in the Wuhan population are healthier than those in New York. This seems unlikely based on estimated life expectancies of the two cities. However, we cannot exclude selection biases [21]. Persons hospitalized in the New York region were skewed towards racial groups and ethnicities known to have poor baseline health and limited health care access for social and financial reasons. However, it is more likely hospital admission criteria were less stringent in Wuhan compared with New York. This bias would also explain several of the more favorable laboratory co-variates and the overall better outcomes in Wuhan.

For example, in Wuhan, beginning February 2020, persons with confirmed COVID-19, even SARS-CoV-2-infection, suspected cases, febrile patients who might be infected would be hospitalized, though the subjects without confirmed COVID-19 were excluded from this study [22]. This differs markedly from the situation in New York where only persons suspected of having moderately severe to severe COVID-19 signs and symptoms were typically hospitalized. For example, nearly two-thirds of cases in the Wuhan cohort had moderate COVID-19 and only 8 died. In contrast, subjects in the New York cohort had to be sufficiently medically ill to be admitted.

Second, hospitalization for COVID-19 in Wuhan was free. This contrasts with most US hospitals where potentially-hospitalized persons face substantial costs which may discourage them from seeking entry unless severely ill [22]. The consequence is it is likely there were fewer less severe cases hospitalized in the Wuhan cohort compared with the New York cohort [23]. Another possibility is mutations in SARS-CoV-2 might result in a more virulent infection and more sever COVID-19 in New York compared with Wuhan. Data supporting this hypothesis are so far lacking.

Differences in infection control policies of the government of China and USA also have an impact on the viral spreading and epidemiology. Chinese government implemented multifaceted interventions to contain the epidemic, such as early detection of cases, contact tracing, population behavioral change (especially mask wearing), social distancing, home quarantine, centralized quarantine, and universal symptom survey [24]. However, infection control policies in the USA in the early period of the pandemic were less stringent and aggressive [25,26,27].

There are important limitations to our study. First, we lacked subject-level data from subjects in the New York cohort and had to rely on published data. Furthermore, we do not know the interval from the onset of symptoms consistent with COVID-19 and hospitalization nor criteria for hospital admission in the New York cohort. Criteria for COVID-19 severity are known for the Wuhan, but not the New York cohort. Moreover, therapy strategies and details may have differed, as well as criteria for hospital discharge. Second, we could not calculate *p*-values for non-normally distributed continuous variables. Third, the pandemic began earlier in Wuhan, therefore, we had final data on all our subjects, whereas only 2634 of 5700 subjects (46%) in the New York cohort had outcome data for comparison.

## 5. Conclusions

We found different hospital admission co-variates and outcomes between hospitalized persons with COVID-19 between Wuhan City and New York. These data should be useful in developing a comprehensive global understanding of the SARS-CoV-2 pandemic and COVID-19.

## Figures and Tables

**Table 1 vaccines-10-00326-t001:** Hospital admission co-variates.

	Wuhan *n* = 1859	New York *n* = 5700	*p*-Value
Age, median (IQR *), years	59 (45, 68)	63 (52, 75)	
Female	925 (50)	2263 (40)	<0.001
Smoker	111 (6)	558 (16)	<0.001
Co-morbidity			
ASCVD *	268 (14)	966 (18)	<0.001
Hypertension	579 (31)	3026 (57)	<0.001
Diabetes	262 (14)	1808 (34)	<0.001
COPD *	61 (3)	287 (5)	<0.001
Cancer	65 (3)	320 (6)	<0.001
Chronic kidney disease	45 (2)	454 (9)	<0.001
Comorbidities			
None	954 (51)	350 (6)	<0.001
1	537 (29)	359 (6)	
>1	368 (20)	4991 (88)	
Temperature > 38 °C	189 (10)	1734 (31)	<0.001
Temperature (°C)	37 (37, 37)	38 (37, 38)	
Laboratory co-variates			
Neutrophils × 10^9^/L	3 (2, 5)	5 (4, 8)	
Lymphocytes × 10^9^/L	1 (0.8, 1.6)	0.88 (0.6, 1.2)	
Lymphocyte < 1000 × 10^9^/L	736 (40)	3387 (60)	<0.001
CRP *, mg/L	13 (3, 51)	130 (64, 269)	
Procalcitonin, ng/mL (<0.5)	0.06 (0.05, 0.1)	0.2 (0.1, 0.6)	
LDH *, U/L (109–245)	212 (170, 292)	404 (300, 552)	
Ferritin, ng/mL (4.6–204)	567 (246, 1218)	798 (411, 1515)	
ALT *, U/L (5–35)	38 (22, 67)	33 (21, 55)	
ALT > 60U/L	533 (29)	2176 (39)	<0.001
AST *, U/L (8–40)	32 (22, 49)	46 (31, 71)	
AST > 40U/L	645 (35)	3263 (58)	<0.001
Creatine kinase, U/L (26–140)	88 (54, 165)	171 (84, 397)	
BNP *, pg/mL (<100)	61 (18, 242)	386 (106, 1997)	
Troponin I above test-specific upper limit of normal	203 (19)	801 (23)	0.006

Data are median (IQR) or n (%). * IQR, interquartile range; * ASCVD, arterio-sclerotic cardio-vascular disease; * COPD, chronic obstructive pulmonary disease; * CRP: C-reactive protein; * LDH: lactate dehydrogenase; * ALT: alanine aminotransferase; * AST: aspartate aminotransferase; * BNP: B-type natriuretic peptide.

**Table 2 vaccines-10-00326-t002:** Hospital course and outcomes.

	Wuhan *n* = 1859	New York *n* = 2634	*p*-Value
Nadir lymphocyte concentration	1.0 (0.6, 1.4)	0.8 (0.5, 1.14)	
IMV *	85 (5)	320 (12)	<0.001
ICU * admission	106 (6)	373 (14)	<0.001
Acute kidney injury	99 (5)	523 (22)	<0.001
Interventions for kidney failure	23 (1)	81 (3)	<0.001
Acute hepatic injury	27 (1)	56 (2)	0.116
Length of stay	18 (12, 23)	4 (2, 7)	
Died	209 (11)	553 (21)	<0.001
no IMV	136/1774 (8)	271/2314 (12)	<0.001
Died, received IMV	73/85 (86)	282/320 (88)	0.576

Data are median (IQR) or *n* (%); * IMV, invasive mechanical ventilation; * ICU, intensive care unit.

**Table 3 vaccines-10-00326-t003:** Deaths by age and sex and hospitalization interval.

	Wuhan Male (%)	New York Male (%)	*p*-Value	Wuhan Female (%)	New York Female (%)	*p*-Value	Wuhan–Hospitalization Median (IQR), d	New York–Hospitalization Median (IQR), d
Age intervals, y								
0–9	0/0	0/13	–	0/0	0/13	–	NA	NA
10–19	0/4	0/1	–	0/1	0/7	–	NA	NA
20–29	1/35 (3)	3/42 (7)	0.621	0/48 (0)	1/55 (2)	1.0	13 (13–13)	4 (1–7)
30–39	2/122 (2)	6/130 (5)	0.283	2/132 (2)	2/81 (3)	0.636	12 (7–24)	3 (2–4)
40–49	2/118 (2)	19/233 (8)	0.016	2/108 (2)	3/119 (3)	1.0	13 (3–43)	6 (3–8)
50–59	25/172 (15)	40/327 (12)	0.486	4/191 (2)	13/188 (7)	0.026	10 (6–18)	6 (3–10)
60––69	42/254 (17)	56/300 (19)	0.577	18/281 (6)	28/233 (12)	0.030	13 (7–20)	6 (3–8)
70–79	52/161 (32)	91/254 (36)	0.525	14/110 (13)	54/197 (27)	0.003	10 (6–19)	5 (3–8)
80–89	27/58 (47)	94/155 (61)	0.087	15/49 (31)	76/158 (48)	0.033	11 (6–19)	4 (2–7)
≥90	3/10 (30)	28/44 (64)	0.078	0/5 (0)	39/84 (46)	0.065	8 (7–NA)	3 (1–6)

## Data Availability

The datasets used and/or analyzed during the current study are available from the corresponding author on reasonable request.

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
