# Peer review of "COVID-19 in China and the US: Differences in Hospital Admission Co-Variates and Outcomes"

_vaccines, 2022, doi:10.3390/vaccines10020326_

Round 1
Reviewer 1 Report
The main aim of this study was to compare the pattern of COVID disease between patients in Wuhan and New York region. The topic is relevant to the field because two dissimilar groups of patients were compared. The authors should consider improving the methodology by comparing the Wuhan patient group with another group with data as detailed as the Wuhan group. The conclusions are consistent with the results, and they address the main question. The references and the tables are appropriate.
Author Response
We agree that the data in New York group should have been as detailed as that in Wuhan group. However, as written in the discussion, one of the limitations of the present study is that detailed data from New York cohort are unavailable. Since the representative roles of these two cities in COVID-19 pandemic, we think it can address the main issues despite the lack of those detailed data.
Reviewer 2 Report
This work is interesting in the way that all the knowledge that we could obtain about COVID-19 can help to improve the outcomes of the patients.
The work is well written and even though the populations compared are very different some conclusions can be obtained. In fact, the authors explain extensively in the discussion the differences in the populations that could explain some of the differences in the results.
Something to improve:
- In the methods section the authors explain details about the Wuhan cohort, but there is not the same explanation about the New York one. Therefore, all the same should be detailed in the text for the New York patients.
Author Response
We are appreciated with this comment and have revised as suggested (Page, Line )
Line 100-104: For the New York cohort, COVID-19 cases confirmed by positive result on PCR testing of a nasopharyngeal sample (the initial test result or if it was negative but repeat testing was positive) and considered to be sufficiently medically ill to require hospital admission were admitted to any of 12 hospitals in Northwell Health, the largest academic health system in New York, between March 1, 2020, and April 4,2020.2.2.
Line 119-123: For the New York cohort, data collected included patient demographic information, comorbidities, home medications, triage vitals, initial laboratory tests, initial electrocardiogram results, diagnoses during the hospital course, inpatient medications, treatments, and outcomes. Data were collected from the enterprise electronic health record (Sunrise Clinical Manager; Allscripts) reporting database.
Line 140-146: For the New York cohort, acute kidney injury was identified as an increase in serum creatinine by 0.3 mg/dL or more (≥26.5 μmol/L) within 48 hours or an increase in serum creatinine to 1.5 times or more baseline within the prior 7 days compared with the preceding 1 year of data in acute care medical records based on the Kidney Disease: Improving Global Outcomes (KDIGO) definition. Acute hepatic injury was defined as an elevation in aspartate aminotransferase or alanine aminotransferase of more than 15 times the upper limit of normal.
Reviewer 3 Report
It is a interesting study for comparison of Wuhan and New York city people who infected with COVID-19 . The authors present the risk factors , demographic data and outcome of the different city and country.
I suggest that the author add the infection control policy of the government of China and U.S.A , which impact the viral spreading and epidemiology.
Author Response
Many thanks for the reviewer’s instructive comment. We have added this point to the discussion.
Line 243-249: Differences in infection control policies of the government of China and U.S.A also have impact on the viral spreading and epidemiology. Chinese government implemented multifaceted interventions to contain the epidemic, such as early detection of cases, contact tracing, population behavioral change(especially mask wearing), social distancing, home quarantine, centralized quarantine, universal symptom survey [24]. However, infection control policies in U.S.A in the early period of the pandemic were less stringent and aggressive [25-27].